# The observation of $\pi$-shifts in the Little-Parks effect in 4Hb-TaS$_2$

Avior Almoalem ●[1], Irena Feldman[1], Ilay Mangel[1], Michael Shlafman[2], Yuval E. Yaish[2], Mark H. Fischer ●[3], Michael Moshe ●[4], Jonathan Ruhman ●[5] & Amit Kanigel ●[1] ✉

Finding evidence of non-trivial pairing states is one of the greatest experimental challenges in the field of unconventional superconductivity. Such evidence requires phase-sensitive probes susceptible to the internal structure of the order parameter. We report the measurement of the Little-Parks effect in the unconventional superconductor candidate 4Hb-TaS$_2$. In half of our rings, which are fabricated from single-crystals, we find a $\pi$-shift in the transition-temperature oscillations. According to theory, such a $\pi$-shift is only possible if the order parameter is non-s-wave. In the absence of crystallographic defects, the shift provides evidence of a multi-component order parameter. Thus, this observation increases the likelihood of the two-component order parameter scenario in 4Hb-TaS$_2$. Furthermore, we show that $T_c$ is enhanced as a function of the out-of-plane field when a constant in-plane field is applied, which we explain using a two-component order-parameter.

The prospect of topological superconductivity showing exotic quantum phenomena such as protected edge states or fractional vortex states with non-abelian statistics has invigorated the field of unconventional superconductivity[1–3]. In order to possess non-trivial bulk topology, the superconductor must be formed out of Cooper-pairs with non-s-wave symmetry. Unfortunately, most superconducting materials favor the mundane s-wave pairing state. While unconventional superconductivity is thus usually associated with correlated electron systems, where strong interactions restrict the pairing channels, understanding the minimal necessary conditions to overcome the natural tendency for conventional superconductivity has proven to be a very difficult task. One of the main challenges is to identify materials that exhibit clear experimental evidence of unconventional superconductivity, where theory and experiment can be carefully compared.

The ability to stack atomically thin materials with different properties and at arbitrary relative angles has revolutionized quantum condensed matter research in the last few years[4]. Such heterostructures show intriguing interacting phases such as superconductivty[5], correlated insulators[6], many-body excitonic states[7], structural and electronic ferroelectrics[8,9], electronic nematicity[10], and magnetism[11,12]. One of the promising prospects of these heterostructures is the ability to reach new electronic ground states not present in any of the constituent layers. An interesting question is whether such heterostructures can be used to manipulate the superconducting pairing channel, resulting in unconventional or even topologically non-trivial superconducting phases.

In this context, the 4Hb polytype of TaS$_2$ provides a particularly interesting example where such a heterostructure is naturally occurring. It constitutes a periodic stack of a Mott insulator and candidate spin liquid[13] (1T-TaS$_2$), and an Ising superconductor (1H-TaS$_2$)[14], which is believed to have an s-wave order parameter[15]. The resulting material forms a highly anisotropic superconductor with $T_c = 2.7$ K. The superconducting state exhibits several unconventional properties, including an enhancement of the muon-spin-relaxation (μSR) rate below $T_c$[16], a residual T-linear specific heat at low temperatures, zero-energy edge states near step edges observed in STM[17] and a mysterious magnetic memory above $T_c$, which manifests itself only as spontaneous vortices observed in the superconducting state[18].

[1]Physics Department, Technion-Israel Institute of Technology, Haifa 32000, Israel. [2]Andrew and Erna Viterbi Faculty of Electrical and Computer Engineering, Technion, Haifa 32000, Israel. [3]Department of Physics, University of Zurich, Winterthurerstrasse 190, 8057 Zurich, Switzerland. [4]Racah Institute of Physics, The Hebrew University of Jerusalem, Jerusalem 91904, Israel. [5]Department of Physics, Bar-Ilan University, 52900 Ramat Gan, Israel. ✉e-mail: amitk@physics.technion.ac.il

The above unconventional properties have led some of us to argue that the order parameter in 4Hb-TaS$_2$ is chiral and possibly topologically non-trivial. Such a state can emerge for order parameters that are degenerate at $T_c$ and can indeed explain the µSR data, the existence of edge states and the spontaneous vortices. However, none of the experiments done so far can provide information about the order parameter and its degeneracy at $T_c$.

In this paper, we provide possible evidence of a two-component order parameter in 4Hb-TaS$_2$ based on the Little–Parks effect[19] in superconducting rings made of single-crystal 4Hb-TaS$_2$. First, we find that in roughly half of the rings, a spontaneous $\pi$-junction is formed, manifested as a $\pi$ shift of the oscillation pattern. Such a shift is direct evidence for an order parameter that changes its sign along the Fermi surface, in other words an unconventional, non-$s$-wave order parameter. Such an effect has previously only been observed in polycrystalline samples[20,21]. Moreover, in the absence of crystallographic defects, the $\pi$-shift requires a two-component order parameter. Second, we find an increase in the critical temperature for out-of-plane fields when a small in-plane field is applied, which can be explained by coupling the out-of-plane field to a chiral order parameter.

## Results and discussion
### Little–Parks effect

The Little–Parks effect is a manifestation of the fluxoid quantization in non-simply connected superconductors. Specifically,

$$\Phi + \frac{mc}{n_s e^2} \oint_C \vec{j}_s \cdot \vec{dl} = n\Phi_0, \tag{1}$$

where $\Phi$ is the magnetic flux penetrating some contour $C$ in the ring, $n_s$ is the superfluid density, $\vec{j}_s$ the supercurrent density and $\Phi_0 = hc/(2e)$ is the flux quantum. The effect is a consequence of the macroscopic nature of the superconducting condensate, which requires the order parameter to be a single-valued function. Therefore, when the applied flux is not an integer in units of $\Phi_0$ the excess flux is screened by a circulating supercurrent, which reduces $T_c$. However, each time $\Phi/\Phi_0 \in \mathbb{Z}$ the full value of $T_c$ is restored and as a result, $T_c$ oscillates as a function of the magnetic field. While for a conventional superconductor $T_c$ has a maximum at zero flux, time-reversal symmetry also allows for a minimum, such that the pattern is shifted by $\pi$. However, such a $\pi$-ring requires a sign-changing order parameter[22].

In practice, it is easier to measure the variation in the resistance of a sample at constant temperature instead of measuring the actual change in $T_c$. For samples that show a sharp superconducting transition, the temperature is stabilized within the transition range. Then, small variations in $T_c$ as a function of the magnetic field lead to a significant change in the resistance. Figure 1a shows a cartoon describing the expected variation in $T_c$ as a function of the magnetic flux through the sample and the corresponding variation of the resistance.

We fabricated several ring-shaped samples (Supplementary Fig. 2) made of 4Hb-TaS$_2$ single crystals (Supplementary Fig. 3) with sizes ranging between $1.2 \times 1.2$ to $0.6 \times 0.6$ µm$^2$ (Supplementary material). Consequently, the expected field periods of the Little–Parks oscillations in the various rings are ~14 to ~60 G. The lateral size of all rings is larger than the coherence length of 4Hb-TaS$_2$ (~35 nm) and of the order of the magnetic penetration depth (~450 nm)[16]. The rings' thickness ranges from ~90 to ~150 nm. Figure 1b shows a scheme of the device and a scanning-electron-microscope image of one of the rings.

Figure 1c shows the temperature dependence of the resistance around the superconducting transition for sample-I. We find two transitions, one at 2.7 K, which we identify with the transition of the large 4Hb-TaS$_2$ pads, and a second transition at a slightly lower

temperature, which corresponds to the ring itself. The width of the latter transition is about 100 mK, similar to the width of the transition measured in a single crystal.

We find Little–Parks oscillations in all the rings with the expected frequency set by the ring dimensions. A typical data set is shown in Fig. 2. Panel (a) shows the resistance as a function of the field for sample-II, a $0.9 \times 0.9$ µm$^2$ ring, at $T = 2.48$ K. The red line is a fourth-order polynomial that is fitted to the data and used for subtracting the background, $R_{bkg} = R_0 + R_2 H^2 + R_4 H^4$. Panel (b) shows the Little–Parks oscillations after the background subtraction, $\Delta R = R - R_{bkg}$.

A crucial step in our experiment is the accurate determination of the zero-field point of the superconducting magnet, which is used to generate the magnetic flux through the ring. To this end, we measure $R(H)$ at a temperature slightly above the bulk $T_c$, where the ring is not fully superconducting but contains superconducting islands that are susceptible to the magnetic field and give rise to a strong magnetoresistance[23]. In Fig. 2c, we show such data for sample-I at $T = 2.65$ K, where no oscillations are observed. The strong magnetoresistance, which is symmetric with respect to zero, allows us to obtain the zero-field point with an accuracy of 1.5 Oe (Supplementary Fig. 6).

Figure 2d shows $\Delta R$ in the temperature range in which oscillations are observed. The amplitude of the oscillations we observe is in good agreement with theory[24]: For an annular ring with inner radius $R_1$ and outer radius $R_2$, the oscillations in the critical temperature are given by $\frac{\Delta T_c}{T_c} = \left(\frac{\xi_0^2}{R_1 R_2}\right)\left(n - \frac{\pi H R_1 R_2}{\Phi_0}\right)^2$. For this ring, sample-I, we estimate $R_1 = 550$ nm, $R_2 = 690$ nm, a coherence length of 35 nm[17], and a critical temperature of 2.7 K to find the largest change in the critical temperature to be $\Delta T_c \approx 2.75$ mK with an oscillation period of 13.5 G. Using the derivative of $R(T)$ at 2.48 K, we find oscillations of $\Delta R \approx 160$ mΩ.

Figure 3 presents the main result of this work. Unlike in the Little–Parks experiments in elemental superconductors, we find two different types of rings, 0-rings and $\pi$-rings. For the 0-ring shown in Fig. 3b, we find a minimum of the resistance at zero-field as expected. However, the $\pi$-rings show a maximum of the resistance at zero-field. This finding provides a clear signature of a half-flux-quantum vortex that is spontaneously created in the $\pi$-rings.

Three out of the eight rings we fabricated show a $\pi$ shift. All rings show either a maximum or minimum at zero field, in other words we never observe a fraction of a $\pi$-shift. The Little–Parks oscillations of all the rings are shown in Supplementary Fig. 7.

The behavior of the different rings is reproducible, meaning that a $\pi$-ring will always show a resistance maximum at zero fields, even for consecutive cooldowns (Supplementary Fig. 8). We have observed the $\pi$-shift in sample-III even after heating to 360 K, well above the CDW transition temperature (see Supplementary material for more details). This strongly suggests that crystal-structure effects play a crucial role in pinning the half-flux vortex.

A half-flux vortex in a superconducting ring can be stabilized in different ways, including by combining superconductors with different order-parameter symmetry or polycrystalline rings[22,25]. In all cases, the involvement of a non-trivial order parameter is a necessary condition to observe this phenomenon. For polycrystalline samples, it is enough to have an order parameter that changes sign under rotation and at least three grains. The half-flux vortex forms when the crystal axes of the grains are rotated with respect to each other in such a way that an odd number of Josephson couplings across the grain boundaries are positive. This results in the frustration of the phase locking between the grains, which is relieved by half-integer flux. A famous example of this effect was measured using a scanning SQUID in the cuprates[20,26,27]: The sample was a ring patterned in a YBCO thin film that was grown on a substrate made of three crystals. These crystals were rotated with respect to each other such that the lobes of the $d$-wave order parameter have a negative overlap across one of the three boundaries.

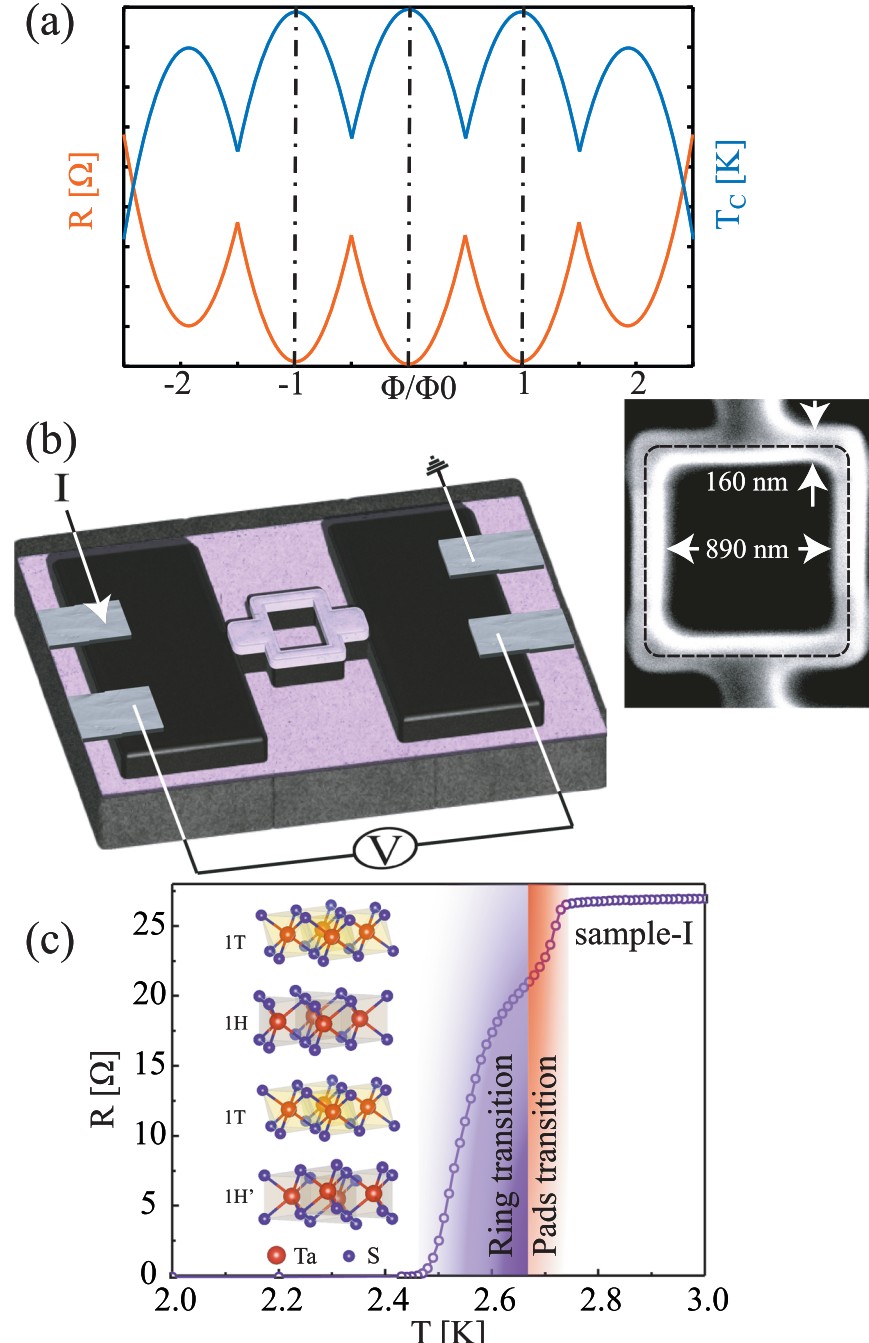

**Fig. 1 | The Little–Parks experiment setup. a** The expected variation of the resistance and of the transition temperature as the flux through the ring is changed. $\Phi$ is the applied flux through the ring, and $\Phi_0$ is the quantum flux. **b** A schematic description of the device. The purple layer represents the $SiO_x$ layer of the substrate and the protective layer. The aluminum contacts are shown in gray, and the black layer is the 4Hb-TaS$_2$ flake. A scanning electron microscope image of a ring is shown. **c** The temperature dependence of the resistance of sample-I, a $1.1 \times 1.1\ \mu m^2$ ring with 140 nm thickness. The colored regimes represent the transitions of the large pads and of the ring, which take place at slightly different temperatures. Inset: The unit cell of 4Hb-TaS$_2$.

The rings in our study, however, are cut from a single crystal and we do not find any evidence for grain boundaries. The ring fabrication process certainly creates structural imperfections, nevertheless, it is very unlikely that a structural tri-junction that hosts a half-flux vortex was spontaneously created in roughly 50% of the rings (see Supplementary materials for a micro-diffraction analysis of a typical ring). Assuming that our rings are indeed predominantly a single crystal greatly restricts the possible order parameters in 4Hb-TaS$_2$, which can lead to a $\pi$-shift.

To understand why, let us first classify the space of order parameters. The crystal structure belongs to the P6$_{3/mmc}$ hexagonal space group (#194). We thus classify the pairing states according to the point group $D_{6h}$ to determine the coupling to external perturbations such as magnetic field or strain. As 4Hb-TaS$_2$ is highly two-dimensional, we restrict ourselves to in-plane pairing, such that we only have to consider four irreducible representations (irreps), namely the two one-dimensional irreps $A_{1g}$ and $B_{1u}$ corresponding to $s$- and $f$-wave pairing, respectively, and the two-dimensional irreps $E_{2g}$ and

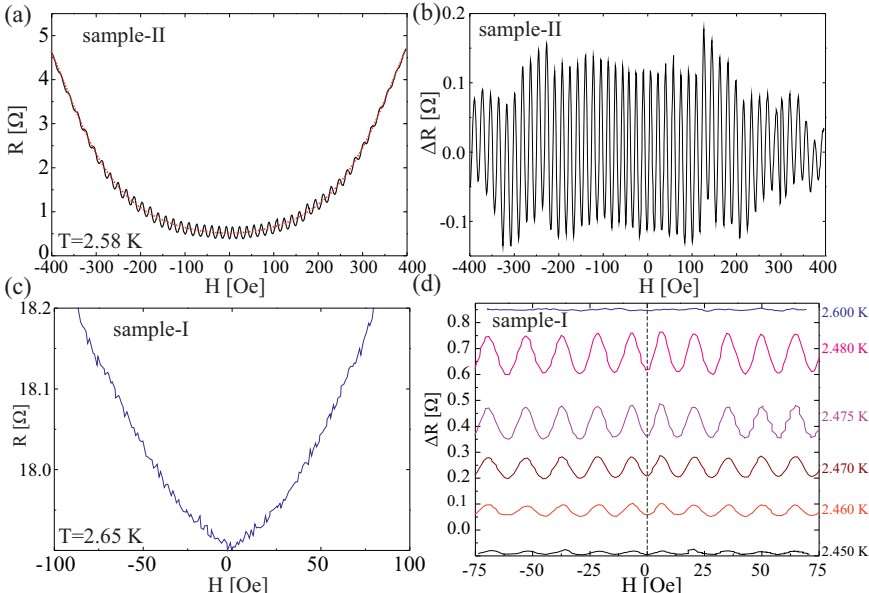

**Fig. 2 | Little–Parks oscillations in 4Hb-TaS$_2$. a** Little–Parks oscillations for sample-II, a ring with a lateral size of $0.9 \times 0.9\ \mu m^2$ and a thickness of 100 nm. The dashed red line is the fourth-order polynomial used for subtracting the background. **b** Same data as in **a** with the background subtracted. As many as 45 oscillations are measured in this ring. **c** Data for sample-I taken at $T = 2.65$ K (see $R$ vs. $T$ for the same sample in Fig. 1c). No oscillations are observed at this temperature. The minimum of the resistance in such curves is used to determine the absolute value of the field in the superconducting magnet. The dashed lines represent the magnetic field at which half of the oscillation period is found in this sample. **d** Temperature dependence of the Little–Parks oscillations from sample-I. The oscillations are observed only in a narrow temperature range.

$E_{1u}$[28]. For the latter two, the relative phase and amplitude between their two components control the time-reversal- and rotational-symmetry breaking of the order parameter. If we denote by $\delta_x$ and $\delta_y$ the real basis of a two-dimensional irrep $E_{1u}$ (These basis functions indeed transform as $x$ and $y$. Note also that the same arguments hold for the case of $E_{2g}$.), a general gap function has the form $\widehat{\Delta}(\theta, \phi) = \Delta_0(\cos\theta\, \delta_x + e^{i\phi}\sin\theta\, \delta_y)\widehat{\sigma}^z(i\widehat{\sigma}^y)$, where $\widehat{\sigma}^i$ are Pauli matrices. Importantly, the spin direction of this spin-triplet gap function is fixed by the strong spin–orbit coupling[14]. For $\theta = \pi/4$ and $\phi = \pm\pi/2$, the order parameter is purely chiral and breaks time-reversal symmetry, while a purely nematic state, which breaks rotation symmetries, is formed by any $\theta$ and $\phi = 0, \pi$.

In what follows, we argue that the frustration required to generate a $\pi$-shift comes from an internal rotation of a two-component order parameter. It is important to caution against this argument. In the presence of crystallographic defects, there exist alternative scenarios that can lead to this effect. Beyond the aforementioned scenario of grain boundaries, it is also well known that local magnetic moments inside a Josephson junction can cause the coupling to change sign[29], thus favouring a $\pi$-shift across the junction. However, since our samples are cut from single crystals, there are no indications of crystallographic defects (see Supplementary data) and no observation of bulk magnetic moments. We explore the two-component order parameter scenario *as the most natural one*, leaving the option of e.g., magnetically or defect-driven $\pi$-shifts to be considered elsewhere.

When the gap belongs to a two-dimensional irrep (in our case $E_{1u}$ or $E_{2g}$), then the superposition of the two components has an internal angle $\theta$ representing the breaking of rotational symmetry, as written above. Similar to polycrystalline samples, a $\pi$ junction will appear if at least three such domains exist and are rotated to the right angles. The question remains, however, what could cause such domains to appear. The experimental observation that $\pi$-rings remain such in successive cool-downs, even when cycling above the CDW temperature, suggests that the origin is structural. Indeed, the two-component order parameter can couple to crystal strain via the free-energy density term[30–32]

$f_{strain} = -\kappa \text{Tr}[\widehat{Q}\widehat{\varepsilon}]$, where $\widehat{\varepsilon}$ is the in-plane strain tensor,

$$\widehat{Q} = |\Delta_0|^2 \begin{pmatrix} \cos 2\theta & \cos\phi\,\sin 2\theta \\ \cos\phi\,\sin 2\theta & -\cos 2\theta \end{pmatrix}, \qquad (2)$$

and $\kappa$ a coupling constant. This term is minimized by a real nematic gap function characterized by $\phi = 0, \pi$, and $\theta$ aligned with the axis of the strain.

Internal strain of the sample can, thus, align the order parameter. Strain is typically smooth and does not form sharp domain walls in a single-crystal. Therefore, we would naively also expect the order parameter to be always smoothly connected around the ring, and as such, free of frustration. However, similar to a classical nematic order parameter in liquid crystals, the strain only defines an axis and not a direction. As such, all planar strain configurations adhere to a topological classification by a winding number $m$, which takes either integer or *half-integer* values[33]. These values represent the number of times the axes of the planar strain field wind while performing a full circuit around the ring. Assuming the two-component order parameter is slaved to the strain field axes, it will find itself internally frustrated when $m$ is half-integer. For example, in Fig. 3c, we schematically plot an $E_u$ order parameter that is slaved to a strain field classified by a winding number $m = 1/2$.

This scenario is consistent with all our observations, namely (i) the $\pi$-shift remains in successive cooldowns because the strain field configuration is likely inherent to the device (e.g. generated by the pads or the substrate). (ii) The $\pi$-shift is observed in only half of our samples. If we assume a random externally driven strain configuration, we may expect $m$ to take random values, thus taking integer and half-integer values with equal probabilities. (iii) Finally, we do not observe a shift by a fraction of $\pi$, which implies the conservation of time-reversal symmetry (TRS). This is naively in contradiction with TRS breaking at low temperatures[16]. However, near $T_c$, the nematic and chiral states are degenerate. Therefore, the presence of strain breaks the degeneracy and prefers the nematic state that conserves TRS. In this scenario, the lower temperature state is a combination of nematic and chiral order

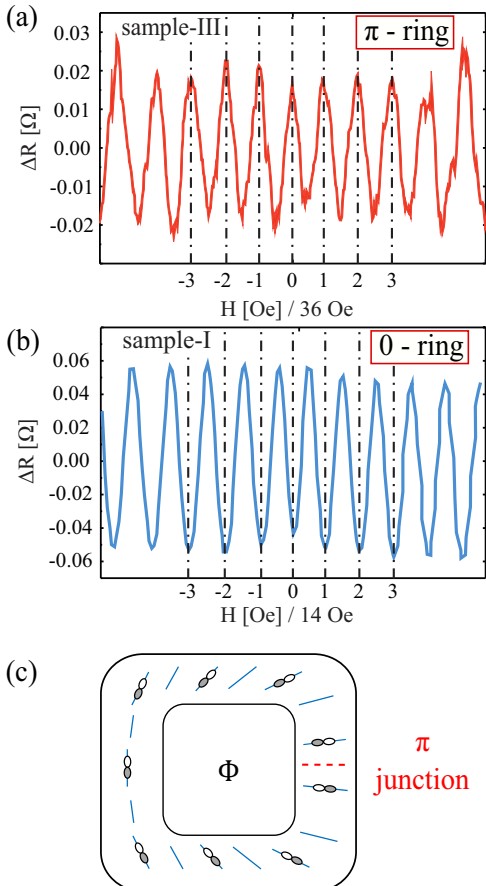

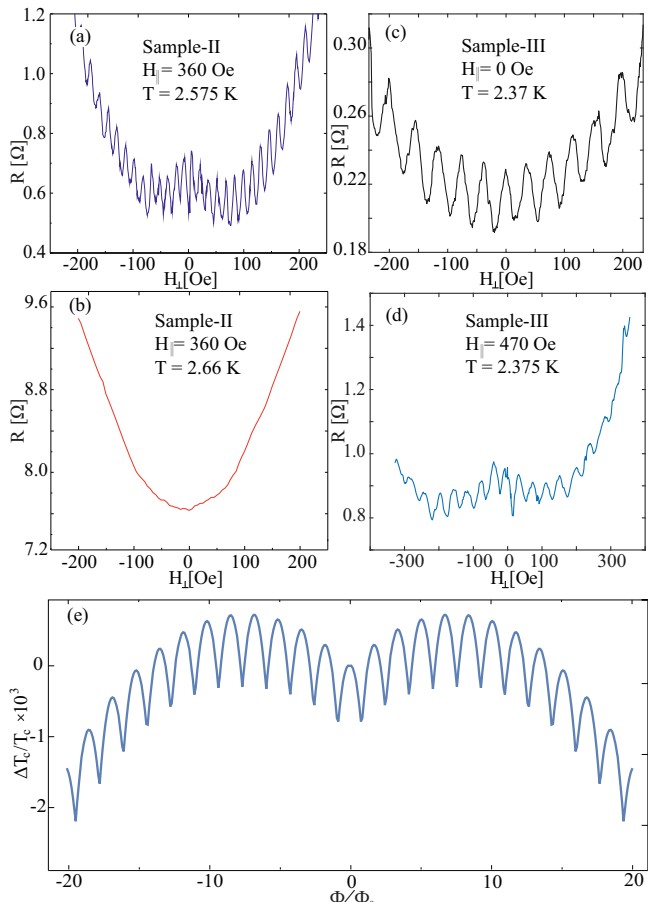

**Fig. 3 | $\pi$-shift in the Little–Parks oscillations. a** The magnetoresistance of a $\pi$-ring. In these rings, the phase of the oscillations is shifted by $\pi$ showing a resistance maximum at zero magnetic field. Data shown was measured at $T = 2.35$ K in a 0.575 $\mu m^2$ ring having a thickness of ~100 nm. **b** The magnetoresistance of a 0-ring, having a minimum at zero magnetic field. In both (**a**) and (**b**), the background was subtracted. **c** Possible explanation for the $\pi$ shift in the Little–Parks oscillations: Strain fields (blue lines) align the two-component order parameter close to $T_c$. The strain field presented here realizes a half vortex. Consequently, the order parameter, schematically represented by the white and gray lobes, can not align with strain without developing a spontaneous $\pi$ junction.

**Fig. 4 | Little–Parks oscillations in the presence of an in-plane magnetic field. a** Same ring as in Fig. 2a, measured in the presence of a 360 Oe in-plane magnetic field. The background has a "Mexican-hat" shape in this case, with minima at about ±80 Oe. **b** Resistance as a function of the out-of-plane field with the 360 Oe in-plane applied at a higher temperature of 2.66 K where the oscillations are absent. The parabolic background found in the absence of the in-plane field, as shown in Fig. 2c, is recovered. **c** Little–Parks oscillations in a $\pi$-ring without in-plane field. **d** Same $\pi$-ring measured in the presence of a 470 Oe in-plane magnetic field. In this case, the background has a minima at about ±150 Oe. **e** Theoretical calculation of $\Delta T_c/T_c$ for an annular ring with similar dimensions and a two-component order parameter. The magnetic field couples to the two-component order parameter linearly and causes the emergence of two maxima. For more details, see Supplementary material.

parameters, minimizing both the strain and the quadratic terms (see Supplementary materials).

## Little–Parks oscillations in the presence of an in-plane magnetic field

For chiral superconductors, it was suggested that in-plane magnetic fields can stabilize the formation of half-flux vortices[34–36]. We thus repeated our Little–Parks measurements in the presence of an in-plane field. For this purpose, a constant in-plane field was applied, and we again measured the resistance as a function of the out-of-plane field. The results for samples II and III are shown in Fig. 4.

Figure 4a shows the resistance as a function of the out-of-plane field in the presence of a fixed 360 Oe in-plane field for sample-II (a 0-ring). While the resistance still shows a minimum at zero field, in other words, the ring remains a 0-ring, the in-plane field clearly modifies the non-oscillatory contribution to the magnetoresistance[37–39], which now exhibits a resistance reduction at small fields with a clear minimum at about 80 Oe. This suggests that a small out-of-plane field now increases the critical temperature. This should be compared with Fig. 2a, which shows the oscillations without the in-plane field for the same sample. In this case, the minimum of the background is clearly at zero field and no negative magnetoresistance can be observed.

In Fig. 4b, we show the resistance as a function of the out-of-plane field with the 360 Oe in-plane field but at a slightly higher temperature, where the Little–Parks oscillations are no longer present. We find a similar field dependence to the one observed without in-plane field. This indicates that this unusual negative magnetoresistance effect is directly related to the onset of a superconducting path around the ring and reflects an increase in $T_c$ with the magnetic field. Finally, in Fig. 4c and d, we show the $T_c$ enhancement in a $\pi$-ring. More details can be found in the Supplementary material.

In general, magnetic fields are known to reduce the transition temperature. Only a few exceptions are known, such as in very thin Pb films, the 2D metallic interfaces LaAlO$_3$/SrTiO$_3$[40], or recently in twisted double bilayer graphene[41] and bilayer graphene[42]. In particular, the coupling of the field to the spins has been suggested to lead to such an effect[43]. Given the strong Ising spin–orbit coupling in this material, such an origin is highly unlikely in this system. In all these examples, an in-plane field leads to an increase in $T_c$. We find a different effect where an out-of-plane field increases $T_c$ in the presence of an in-plane field.

A more plausible explanation here is the orbital coupling between a chiral order parameter and a magnetic field $f_{chiral} = -K_c H |\Delta_0|^2 \sin 2\theta \sin \phi$. Such a term favors a chiral state $\theta = \pi/4$ and $\phi = \pm \pi/2$ enhancing $T_c$ linearly with the absolute value of the magnetic field and only appears for two-component order parameters (for details see Supplementary material). In Fig. 4e, we plot a theoretically calculated Little–Parks oscillation pattern with such a term, which is in good qualitative agreement with the data.

To conclude, by measuring the Little–Parks effect, we provide convincing evidence that the superconducting order parameter in 4Hb-TaS$_2$ belongs to a two-component representation. In particular, we present two main findings, both of which are most easily explained by a two-component order parameter. First, The Little–Parks oscillations of roughly half of our rings exhibit $\pi$-shift in spite of the fact that the rings are formed out of a single crystal. Second, the envelope of the resistance oscillations shows a clear minimum at a finite field (for a constant 360 Oe in-plane field), indicating an increase in $T_c$ for finite applied fields.

We propose that the $\pi$-shift arises due to topological configurations of the planar strain field in the ring. This strain field couples to the two-component order parameter and causes it to form and rotate as it goes around the ring. However, because the strain axe is like a nematic director, it also supports half vertices, which naturally frustrates the two-component order parameter when it is slaved to the strain. The origin of such strain is not currently known. It may result from the sample growth procedure, which essentially includes a rapid cool down, or the stresses applied on the ring by the pads and substrate due to thermal compression in the cooling. We also do not have an explanation for the appearance of half-vortices of strain in half the samples. But this naturally arises if we assume the topological winding is a random number.

Finally, we have argued that the increase in $T_c$ as a function of a perpendicular magnetic field indicates a chiral component, which is induced by the field itself. However, the fact that this $T_c$ enhancement is only seen with an in-plane field is not naturally explained by this scenario. In the Supplementary material, we show that when both strain and field are present, the order parameter is a mixture of chiral and nematic states with $\phi = \pi/2$ and $0 < \theta < \pi/4$. In this scenario, the enhancement of $T_c$ with the field is suppressed due to the competition with strain. Thus, we may speculate that if the coupling to strain is suppressed by the in-plane field, it will lead to the observed $T_c$ enhancement.

The consistent picture thus emerging from our experiments is the following: Right at $T_c$, 4Hb-TaS$_2$ enters a superconducting phase that belongs to a two-dimensional irreducible representation. Close to $T_c$ this degeneracy is lifted by internal strain in the sample, which can lead to both 0- and $\pi$-rings. Only upon lowering the temperature further does the system break time-reversal symmetry spontaneously by forming a chiral order parameter. We note that recent measurements of an anisotropic in-plane $H_{c2}$ also support this scenario, where it was argued that the order parameter is a mixture of chiral and nematic states tuned by temperature[44]. The microscopic origin of the two-component order parameter thus remains an outstanding open question.

## Methods
### Sample preparation
High-quality single crystals of 4Hb-TaS$_2$ were grown using the chemical vapor transport (CVT) method. A stoichiometric mixture of tantalum (Ta) and sulfur (S) was sealed in a quartz ampoule under vacuum. 1% of Se was added to the mixture. It was found to significantly improve the sample quality. The mixture underwent a sintering process, forming a boule of 4Hb-TaS$_{1.99}$Se$_{0.01}$. The boule was crushed and placed in a 200-mm-long quartz ampoule with a 16-mm diameter. Iodine was added as

a transport agent, and the ampoule was sealed under vacuum. The ampoule was then placed in a three-zone furnace, where the hot ends were heated to 800 °C, and the middle part was kept at 750 °C. After about 30 days, the ampoule was quenched in cold water. Crystals having a few mm in size are found in the cold part of the ampoule.

The crystal structure and chemical composition were verified using X-ray diffraction and electron energy dispersive spectroscopy in a scanning electron microscope. The measurements show that the actual amount of Se in the crystals is <1%.

### Rings fabrication
For this experiment, we fabricated 4Hb-TaS$_2$ rings with typical dimensions of less than 1 μm$^2$. We exfoliate 4Hb-TaS flakes on a SiO$_2$/Si substrate using the standard dry transfer technique. Four Ti/Al contacts are evaporated using electron beam lithography. We cover the flakes with a ~500 nm protective layer of SiO$_2$ in-situ before using a FEI Helios NanoLab DualBeam G3 UC focused ion beam (FIB) to carve the desired ring from the exfoliated flake. For carving we use a 30 kV, 40 pA current.

### Magneto-resistance measurements
The magneto-resistance measurements were done using a Quantum-Design DynaCool system with temperature stability of ±180 μK. The resistance was measured by applying an AC current and measuring the voltage in 4 contacts configuration. All the rings have shown Ohmic behavior in the entire range of temperature and magnetic field, in agreement with measurements in different systems[21]. We compare the Little–Parks oscillations measured with a current of 300 and 800 nA (Supplementary material). All the data presented in the paper was measured with a current of less than 300 nA.

### Finding the zero-field in the superconducting magnet
We use a superconducting magnet, trapped flux makes it difficult to find the "real" zero-field.

Before each experiment, we minimize the trapped flux by oscillating the magnetic field to +1 T, −0.5 T, +1 kG, −500 G, +100 G and finally 0.

We first stabilize the temperature within the superconducting transition so that it shows a strong magneto-resistance but no oscillations (see Fig. 2c). We then measure $R(B)$ for a range of magnetic field around "zero". The zero-filed is set to be the field at which the resistance is minimal.

We found that as long as the magnetic fields are smaller than 400 G, the residual field does not change and remains smaller than 5 G. We measure the residual magnetic field in the system before and after each Little–Parks measurement.

We emphasize that our uncertainty in the size of the magnetic field is smaller than half of the Little–Parks oscillation period for all the samples that we measured.

### In-plane magnetic field
The quantum design PPMS system allows only the application of magnetic fields normal to the sample. To add an in-plane magnetic field we built a Helmholtz coil that is inserted into the PPMS sample space. The coil was designed so that the sample is located at its center while inside the PPMS system. The apparatus is made out of two identical coils with a radius of 10 mm. Each coil contains 200 turns of a superconducting Sn–Nb wire, with a critical temperature of 7 K, around an aluminum core. We built the apparatus such that the centers of the coils are 10 mm away. We calibrated the coils outside of the PPMS system using a Tesla-meter to find a ratio of 180 G per 1 A. This was compared to a simulation giving the same results. Inside the PPMS, due to the cooling-power limit of the system, the coils can provide an in-plane field of up to 500 G.

## Data availability

All the data that support the findings of this study are provided in the main text and the Supplementary Information.

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

## Acknowledgements

We thank Jorn Venderbos, Patrick Lee, Avraham Klein, Daniel Agterberg, and Erez Berg for very helpful discussions. We thank Itay Mangel, Noa Somech, and Shay Hacohen-Gourgy for their help with the experiment. We thank Alex Brener for his help with the EBSD measurements. J.R. was supported by the Israeli Science Foundation grant no. ISF-3467/21. A.A., I.F., and A.K. were supported by the Israeli Science Foundation grant no. ISF-1263/21.

## Author contributions

A.A. and A.K. devised the experiment. I.F. and A.K. grow the samples. A.A. fabricated the devices with the help of M.S. and Y.Y. A.A. performed the measurements. I.M. performed the EBSD experiment. M.F., M.M., and J.R. did the theoretical calculations. A.A., J.R. and A.K. wrote the manuscript.

## Competing interests

The authors declare no competing interests.
