## [Peer Review File · Nature Communications]

The observation of π -shifts in the Little-Parks effect in 4Hb-TaS₂REVIEWER COMMENTS

Reviewer 1 (Remarks to the Author):

In the submitted manuscript ‘*Evidence of a two-component order parameter in 4Hb – TaS₂ in the Little-Parks effect*’ the authors present results from Little-Parks experiments on ring-shaped single-crystal 4Hb – TaS₂ samples where they find evidence for the superconducting order parameter to be constructed from two condensates. More specifically, the authors observe two effects which are seemingly at odds with a single-component scenario, but naturally explained through a two-component superconductor. (1) They find an increase in T_c at small out-of-plane magnetic fields H_z , given that an in-plane magnetic field \mathbf{H}_{\parallel} is applied. (2) In almost half of their samples, they observe a π -shift in the oscillatory behavior which they attribute to a half-flux vortex caused by a rotation of the nematic director across the ring.

The presentation of the results is clear and detailed, and the data interpretation is done with good scientific care. Moreover, the findings add meaningful evidence to the puzzle on the superconducting pairing state in 4Hb – TaS₂ and are probably of interest for a broader physics audience. For these reasons, I recommend a publication of the submitted manuscript in Nature Communications. In the following I will list a couple of questions and notes, that I would appreciate to see addressed by the authors.

Remarks and questions:

- In your theory part, is it possible to extend your derivation to the case where a half-flux quantum is present ? I.e. can you derive a similar expression for the π -shifted parabola ? From what I understand, the strain pattern is much more subtle, but maybe for the simplified idea sketched in Fig. 3(c).
- Can one draw a little sketch of your proposed superconducting scenario, as a quick summary? [First, nematic, then TRSB mixing, etc.]
- Out of curiosity, I redid your derivation starting Eq. (2) in the SM, but kept the magnetic field direction arbitrary. In case, all previous assumptions go through one ends up with a term under the square-root where the suppression of strain crucially depends on an expression $[1 + \cos(2\varphi_H - \theta_E)]$ with the in-plane angle φ_H of the magnetic field and the angle θ_E of the in-plane strain.

I understand your current apparatus does not allow for in-plane rotation of the magnetic field, but would you find it plausible that the T_c enhancement would go away by changing φ_H [maybe rather in 0-samples where the chance of strain being actually homogeneous is higher] ?

Also, in SM-Eq. (3), I believe the second term should have a prefactor $(K_1 - K_2)$, then also g and g' become essentially the same expressions.

Also, do you find the T_c enhancement in all of your samples ?

- “*As a consequence it will vanish near this region and develop a spontaneous π junction.*”
Is that so obvious ? Why could the order parameter not connect by keeping the magnitude and quickly sweeping over the complex plain ?
- “*When a π flux is introduced to the center hole this frustration is removed, thus causing T_c to increase.*”
Could you elaborate more on that frustration to make it easier to understand ?
- “*The state right below T_c is then a “nematic-chiral” mixture with $\phi = \pm\pi/2$ (depending on the direction of the field).*”

Just to be clear, the relative phase is only $\phi = \pm\pi/2$ for as long as only strain $\epsilon_{xx} - \epsilon_{yy}$ is present. As soon as ϵ_{xy} enters, the phase is anything $\phi \neq \{0, \pm\pi/2\}$. Right ?

- I believe it is semantics but in your SM- Sec. “The chiral-nematic mixed state in the presence of strain and magnetic field”, you focus quite a bit in restoring the linear behavior. You equally get your maxima if the total curvature at $\Phi = 0$ is positive. In other words, the leading term from the square-root being quadratic is not a problem.
- How is $\boldsymbol{\sigma} \cdot \mathbf{n} = 0$ to be understood, given that $\boldsymbol{\sigma}$ is a tensor?
- In your first [Little-Parks] equation, please specify what Φ' is, and also later or in SM how ΔT_c is defined.

Reviewer #2 (Remarks to the Author):

Dear Authors,

Please find my review of the manuscript "Evidence of a two-component order parameter in 4Hb-TaS₂ in the Little-Parks effect".

This is an experimental magneto-resistance study examining the Little-Parks oscillations in micron-sized rings of tantalum disulfide (4Hb polytype) patterned by FIB from exfoliated flakes.

Intriguingly, in 3 out of 8 rings, the authors find that the oscillations are shifted by a factor of π , which provides clear evidence of a half-flux-quantum vortex. This effect was reproduced in all of the cool down cycles performed by the authors. Such a π -shift in Little Parks oscillations necessarily implies "the presence of a non-trivial (non s-wave) order parameter". The authors also find a T_c increase with small in-plane field.

The authors provide data about all 8 rings, with ample details in the supplementary, and I did not find anything to criticize about the data. So up to that point I concur with the authors, and I think this result is worthy enough of publications in and of itself: there are only few compounds in which this has been found and they are all quite special (cuprates, Sr₂RuO₄, Bi₂Pd). Such a π -shift is rather unusual, and the well-known dichalcogenide family of compounds are usually thought of as standard s-wave superconductors, so this is clearly of significance to the field of dichalcogenides and more generally of superconductivity.

Now, I disagree with the claim that this is enough evidence "of a two-component order parameter".

The scenario proposed by the authors based on strain is coherent, but it is just a scenario, there is no direct evidence to support it, only indirect evidence. Indeed, it all rests on the affirmation that "The rings in our study, however, are cut from a single crystal and we do not find any evidence for grain boundaries." Some kind of microdiffraction measurements of local strain could maybe provide evidence but it is not in the manuscript.

And at the same time the proposed scenario based on strain has some caveats, for instance:

-The authors themselves also recognize at the end that for their proposed scenario: "However, the fact that this T_c enhancement is only seen with an inplane field is not naturally explained by this scenario."

-In addition, "The origin of such dislocation defects is not currently known. [...] It is also not clear why the defects occur in roughly half of the devices." But such a ratio is straightforward in the polycrystalline scenario that the authors dismissed (assuming there is a random number of π -junctions in the device, that random number can be even or odd, hence a ratio of π -ring to 0-ring that is close to 50/50).

By comparison, there is a similar finding in Ref[21] Science 366, 238 (2019) and its supplementary materials, where Y. Li, et al. also find a π shift in the Little-Parks oscillations of rings of Bi₂Pd. However, Y.Li et al. go into much more specific evidence: they find a ratio of eleven π -rings to three 0-rings (or 13 to 8, including some failed devices) only in polycrystalline samples. While in epitaxial sample (with no grain boundaries) they do not find π -rings. They also back it up with TEM cross-section results showing grain boundaries and the size of the grains. Thus, they can quite conclusively attribute the π -shift/half quantum vortex to π -junctions at grain boundaries (thus supporting a p-wave SC scenario). In comparison, I find that the current manuscript is lacking such a similar microstructural study to really back up its claim of a "two-component order parameter".

Another thing that gives me doubts is the two-step superconducting transition. I understand it could be due to strain, but, for such a tiny samples, how do you rule out intercalation/contamination by Gallium ions from the FIB cuts ? or maybe even self-

doping/intercalation from redeposited materials from the FIB cuts ? or indeed a polycrystalline scenario ? Here comparison with rings fabricated by etching with a noble gas (Helium FIB or Argon ion milling,...) would provide more conclusive evidence.

Also, in the dichalcogenide family, and TaS₂ in particular, there can be some strong effects of disorder on the superconducting T_c (both up or down) due to the competition with the CDW. What if the change in T_c is due to disorder created by the Ga-ion beam or the electron beam of the microscope?

Finally, some of the coauthors of this manuscript actually disagree too that this is definitive evidence "of a two-component order parameter". Indeed they provide an alternative explanation for the pi-shift in ref. M. H. Fischer, P. A. Lee, J. Ruhman arXiv:2304.10583v2 where they even explicitly state so: "[...] this paper offers an alternative explanation of the pi-phase shift to that given in the original paper [3] ...". So, at least, to ensure self-consistency, the manuscript should be updated to reflect the fact that some of the coauthors proposed an alternative explanation in arXiv:2304.10583v2. I understand this alternative explanation came after the original manuscript was written, but it does rebut the main claim of the title so it cannot stay as is.

Indeed, the title "Evidence of a two-component OP ..." seems too strong as it is only one possible scenario. "Evidence of half-quantum flux" or "evidence of n-shift in the Little-Parks oscillations" seems more likely to stand the test of time (the current title appears to have already lost that test), all the while being peculiar enough to warrant publication.

So, I would indeed be willing to accept this manuscript, provided the authors strongly requalify their claims, as mentioned above.

Reviewer #3 (Remarks to the Author):

Almoalem et al. report unusual behaviors in the Little-Parks effect of nanofabricated superconducting 4Hb-TaS₂ rings. First, they find that some of their rings show Little-Parks oscillations with a peak in resistance at zero magnetic field, instead of a dip as in the conventional case. Such a pi-shift of the Little-Parks oscillations was previously seen only in a polycrystalline Bi₂Pd ring. Bi₂Pd is a candidate material for topological superconductivity with unconventional pairing and the randomly distributed grains in a polycrystalline ring can allow the occurrence of odd number of pi-junctions. It is in this setting that the observation of a pi-shift is quite unexpected in a single crystalline ring. The authors proposed that strain in such a single crystal could give rise to an effective "grain boundary". Second, the authors report that upon applying a small in-plane magnetic field, the resistance of the samples at certain temperatures first decreases and then increases with the perpendicular magnetic field. The authors attribute this decrease to enhanced superconductivity and relate this phenomenon to chiral superconductivity.

While both of the reported experimental features are interesting, the paper seems to lack a comprehensive and in-depth investigation of them. The important claims such as two-component superconductivity and chiral superconductivity are based on a few traces at a handful of temperature points or in-plane magnetic fields. The authors swiftly ignored other (mundane) origins and jumped directly to the exotic one. The manuscript fails to meet the high standard of Nature Communications and I cannot recommend for publication. I list more detailed reasons below:

1. The authors use focused ion beam to fabricate all their rings. This technique typically creates an amorphous layer of about 10-20 nm under grazing incidence polishing [P. Moll, Annu. Rev. Condens. Matter Phys. 9, 147 (2018)]. Both the inner and outer sides of the ring after FIB would host such amorphous layers. Since the thickness of the ring—the difference between the inner and outer radii—is about 100 nm, that means about 20-40% of the ring consists of damaged layers. This damaged layer seems to account for the reduced superconducting transition temperature with

a broader transition width, as shown in Fig. 1 and Fig. S4. The presence of a large portion of damaged layers speaks against the authors' claim that their sample is fully single crystalline.

The amorphous layer may consist of polycrystalline domains. These domains can form an odd number of grain boundaries, similar to a tri-crystal of d-wave superconductors. This scenario could explain the occurrence of pi-junctions.

In fact, some latest experiments showed that the influence of FIB on two-dimensional materials is much deeper and subtler [F. Sarcan, et al., npj 2D Mater. Appl. 7, 23 (2023)]. As a suggestion, the author may consider using standard e-beam lithography and reactive ion etching to avoid the damage brought by FIB.

2. The authors spend a substantial portion of their analysis, including Fig. 1 and Fig. 2, on the Little-Parks oscillations of a conventional ring. The readers can only find a limited set of data from the pi-ring (To be exact, just one trace in the main text). Often the readers have to delve into the supplementary information. The authors need to clarify the following issues:

a. Have the authors checked the temperature dependence of ΔR in the pi-rings? Does it agree with that expected from the formula of Little-Parks effect? It is not convincing enough, as is done in the current version, to compare ΔR from experiment and theory at a single temperature point. The authors should present the experimental and theoretical curves of ΔR as a function of T . They should also provide the complete data set of Little-Parks oscillations in all the pi-rings at different temperatures.

b. Apart from the temperature dependence, do the oscillation period agree with the ring geometry in all their rings? It seems that the third pi-ring [Fig. S4(8)] has some double-peak structure around ± 40 Oe. What is the reason for that?

c. How do the pi-rings look like? The authors should provide SEM images for all their rings.

d. Do the authors calibrate zero field in the same way for all the pi-rings? If so, they should present the data.

3. The scenario of some kind of grain boundary formed by strain is highly speculative. I have several doubts on this proposal:

a. it is understood that FIB patterned structures may experience strain. However, that happens for samples that are glued to the substrate [P. Moll, Annu. Rev. Condens. Matter Phys. 9, 147 (2018)]. Here the sample is exfoliated on the substrate and the bonding between the flake and the substrate is van der Waals bonding. It seems unlikely that strain can be created.

b. The authors show no experimental evidence for the strain effect. In principle, if it is due to strain, there are several ways to check this. For instance, one could do further patterning on the pi-ring to check if it becomes a 0-ring due to the release of strain. A more desirable experiment would be to tune the strain in-situ.

4. The authors argue that enhanced superconductivity with an in-plane magnetic field stems from chiral superconductivity. In principle, this effect should apply to a bulk superconductor of 4Hb-TaS₂ as well. Have the authors checked this?

5. Related to the question above, the authors mentioned that their initial intention to introduce the in-plane magnetic field is to stabilize the pi-junction because theory expects that a chiral superconductor should host this property. However, it seems that they fail to realize this behavior. This negative result seems to question the validity of chiral superconductivity in this system.

6. The authors presented only two traces in the main text for the decrease of resistance with increasing magnetic field [Fig. 4(a), Fig. 4(d)]. How does this behavior depend on temperature, as long as the ring becomes superconducting? In fact, the observation that such a w-shaped curve only occurs if the ring becomes superconducting is worrisome. It indicates again that it has something to do with the damaged surface after FIB, as I explained in the first point.

Furthermore, how does the behavior evolve with the in-plane magnetic field? For the two traces in Fig. 4(a) and Fig. 4(d), the authors used different in-plane magnetic fields. Why don't the authors study this behavior in a single sample but at a wide range of in-plane magnetic fields?

7. A home-built coil was used for generating the in-plane magnetic field. It is necessary to explain in more details how the coils are built: diameter of the coils, distance between the coils, number of turns and layers, thickness of the wire, calibration of the field, simulation of the homogeneity of the field, etc.

8. The authors argued that the enhancement of superconductivity by an in-plane magnetic field in the previous cases was different because 2H-TaS₂ has strong Ising superconductivity. In principle, Pb has a very strong spin-orbit coupling too. This is evidenced by its large in-plane upper critical field [T. Sekihara, et al., Phys. Rev. Lett. 111, 057005 (2013)], similar to Ising superconductivity. As the authors mentioned, enhanced superconductivity by an in-plane field was observed in Pb films too. It seems that it is not necessary to involve chiral superconductivity to see this effect. In general, the enhanced superconductivity by an in-plane magnetic field can have multiple origins and remains not fully understood.

9. What are the thicknesses of the exfoliated flakes used here?

10. I recommend the authors to use H_{\perp} instead of H to label the x-axis. Because H may be interpreted as the total magnetic field, instead of the out-of-plane field.

11. The paper seems to be prepared in a hasty manner. Most of the sections in the supplementary information are merely repeating the same texts in the method section. The page number of ref. 42 is wrong.

Dear Referees,

We thank you for your careful reading of our paper and your suggestions. In the enclosed document we respond point-by-point to your comments.

Sincerely,

Amit Kanigel on behalf of all authors

Color coding:

Black - original Referee report

Blue - response to Referees

Red - quoted text from the manuscript

Report of Referee A

In the submitted manuscript 'Evidence of a two-component order parameter in 4Hb – TaS₂ in the Little-Parks effect' the authors present results from Little-Parks experiments on ring-shaped single crystal 4Hb – TaS₂ samples where they find evidence for the superconducting order parameter to be constructed from two condensates. More specifically, the authors observe two effects which are seemingly at odds with a single-component scenario, but naturally explained through a two-component superconductor. (1) They find an increase in T_c at small out-of-plane magnetic fields H_z , given that an in-plane magnetic field H_k is applied. (2) In almost half of their samples, they observe a π -shift in the oscillatory behavior which they attribute to a half-flux vortex caused by a rotation of the nematic director across the ring. The presentation of the results is clear and detailed, and the data interpretation is done with good scientific care. Moreover, the findings add meaningful evidence to the puzzle on the superconducting pairing state in 4Hb – TaS₂ and are probably of interest for a broader physics audience. For these reasons, I recommend a publication of the submitted manuscript in Nature Communications.

We thank the Referee for carefully reading our paper and for the detailed and technical comments.

In the following I will list a couple of questions and notes that I would appreciate to see addressed by the authors.

Remarks and questions:

- In your theory part, is it possible to extend your derivation to the case where a half-flux quantum is present ? I.e. can you derive a similar expression for the π -shifted parabola ? From what I understand, the strain pattern is much more subtle, but maybe for the simplified idea sketched in Fig. 3(c).

Yes, it is possible. In the revised version of the supplementary materials we add a detailed analysis of this case and show that a strain driven half-vortex of the two-component order parameter induces a π -shift of the LP oscillations.

- Can one draw a little sketch of your proposed superconducting scenario, as a quick summary? [First, nematic, then TRSB mixing, etc.]

Yes, we add a sketch taken from arxiv 2208.14442. The schematic plot shows the magnitude of the order parameter $\eta = (\cos \theta + e^{i\phi} \sin \theta) \Delta_0$. The colorbar indicates the value of the angle ϕ (where $\phi = 0$ is purely nematic and $\phi = \pi/4$ is chiral). We assume that the low-temperature phase $T \ll T_c$ is predominantly a chiral superconductor because the phenomenology in 4Hb-TaS2 is consistent with a fully gapped superconductor. In the language of the GL theory Eq.2 this implies that the quartic coefficient β_2 is negative, which naturally favors a

chiral state in a manner that breaks TRS spontaneously. However, magnetic fields and strain are coupled to the order parameter as well, and may alter this picture. Importantly, these couplings are quadratic rather than quartic. Consequently, close enough to T_c the quadratic terms are dominant over the quartic term and dictate the nature of the order parameter.

The picture we have in mind is that near T_c the SC order parameter is predominantly nematic due to strain. As the temperature is lowered the quartic term becomes more and more important leading to a mixture of nematic and chiral states. However, we assume that this mixture becomes predominantly chiral very quickly to fit the phenomenology of the SC. This is obtained in the limit $\Delta_0^2 \kappa \ll \beta_2$, where Δ_0 is the order parameter at zero temperature.

In the conclusion section we explain this where we write:

“The consistent picture thus emerging from our experiments is the following: Right at T_c , 4Hb-TaS2 enters a superconducting phase that belongs to a two-dimensional irreducible representation. Close to T_c this degeneracy is lifted by internal strain in the sample, which can lead to both 0- and π -rings. Only upon lowering the temperature further, the system breaks time-reversal symmetry spontaneously by forming a chiral order parameter. We note that recent measurements of an anisotropic in plane H_{c2} also support this scenario, where it was argued that the order parameter is a mixture of chiral and nematic state tuned by temperature [46]. The microscopic origin of the two-component order parameter thus remains an outstanding open question.

”

- Out of curiosity, I redid your derivation starting Eq. (2) in the SM, but kept the magnetic field direction arbitrary. In case, all previous assumptions go through one ends up with a term under the square-root where the suppression of strain crucially depends on an expression $[1 + \cos(2\varphi_H - \theta_E)]$ with the in-plane angle φ_H of the magnetic field and the angle θ_E of the in-plane strain. I understand your current apparatus does not allow for in-plane rotation of the magnetic field, but would you

find it plausible that the T_c enhancement would go away by changing φ_H [maybe rather in 0-samples where the chance of strain being actually homogeneous is higher] ? Also, in SM-Eq. (3), I believe the second term should have a prefactor ($K1 - K2$), then also g and g_0 become essentially the same expressions. Also, do you find the T_c enhancement in all of your samples ?

Regarding the arbitrary direction of the magnetic field. We are not sure we understand the comment made by the referee. The GL theory we consider is two-dimensional. Except for the effects on the band structure, the superconducting order parameter (in the E_u or E_g irreps) couples only to out of plane magnetic fields. Thus, an in plane magnetic field will not appear in Eq. (2) and will not lead to a term that depends on φ_H .

Regarding the comment about the algebraic mistake. We agree with the referee, the coefficient in front of the derivative term along y is indeed $K1-K2$. In the revised version of the supplement we have corrected this error.

- “As a consequence it will vanish near this region and develop a spontaneous junction.” Is that so obvious ? Why could the order parameter not connect by keeping the magnitude and quickly sweeping over the complex plain ?

We apologize for the incorrect and confusing statement. In the revised manuscript we show that the minimum energy solution is that of a half vortex and not characterized by a spontaneous π -Josephson junction. Namely, we add a half-integer phase winding to the order parameter to compensate for the frustration of the two-component order parameter. This solution is energetically preferable in the limit $R_2 - R_1 \gg \pi\xi \log R_2/R_1$, where the cost of a half vortex is smaller than a domain wall of width ξ .

- “When a flux is introduced to the center hole this frustration is removed, thus causing T_c to increase.” Could you elaborate more on that frustration to make it easier to understand ?

See our response to the two previous comments. We have now revised the supplement and these confusing sentences were removed. Instead there is a rigorous analysis.

- “The state right below T_c is then a “nematic-chiral” mixture with $\phi = \pm \pi/2$ (depending on the direction of the field)”. i Just to be clear, the relative phase is only $\phi = \pm \pi/2$ for as long as only strain $xx-yy$ is present. As soon as xy enters, the phase is anything $\phi \neq \{0, \pm \pi/2\}$. Right ?

Yes. To make this clearer we now changed the text in the supplement to be

“...depending on the direction of the strain field”

- I believe it is semantics but in your SM- Sec. “The chiral-nematic mixed state in the presence of strain and magnetic field”, you focus quite a bit in restoring the linear behavior. You equally get your maxima if the total curvature at $\Phi = 0$ is positive. In other words, the leading term from the square-root being quadratic is not a problem.

We agree. The plots were prepared for the linear case and so we discuss the limitations of these plots, but we agree that even outside the regime of validity of this approximation the behavior seen in experiment, where T_c goes up with field close to zero, is possible (depending on parameters)

- How is $n \cdot \sigma = 0$ to be understood, given that σ is a tensor?

The notation should be regarded as $\sum_j \sigma_{ij} n_j = 0$. We now write it explicitly.

- In your first [Little-Parks] equation, please specify what Φ' is, and also later or in SM how ΔT_c is defined. ii

We have erased Φ' from the equation. It was supposed to refer to the difference between the gauge invariant current and the flux term. ΔT_c is not sharply defined. It refers to the quadratic envelope of the T_c curve (between the point at zero flux and 10 flux quanta) and is only used in free speech to argue there is even rough quantitative agreement between theory and experiment.

Report of Referee B:

Dear Editor and Authors,

Please find my review of the manuscript “Evidence of a two-component order parameter in 4Hb-TaS₂ in the Little-Parks effect”.

This is an experimental magneto-resistance study examining the Little-Parks oscillations in micron-sized rings of tantalum disulfide (4Hb polytype) patterned by FIB from exfoliated flakes.

Intriguingly, in 3 out of 8 rings, the authors find that the oscillations are shifted by a factor of π , which provides clear evidence of a half-flux-quantum vortex. This effect was reproduced in all of the cool down cycles performed by the authors. Such a π -shift in Little Parks oscillations necessarily implies “the presence of a non-trivial (non s-wave) order parameter”. The authors also find a T_c increase with small in-plane field.

The authors provide data about all 8 rings, with ample details in the supplementary, and I did not find anything to criticize about the data. So up to that point I concur with the authors, and I think this result is worthy enough of publications in and of itself: there are only few compounds in which this has been found and they are all quite special (cuprates, Sr₂RuO₄, Bi₂Pd). Such a π -shift is rather unusual, and the well-known dichalcogenide family of compounds are usually thought of as standard s-wave superconductors, so this is clearly of significance to the field of dichalcogenides and more generally of superconductivity.

Now, I disagree with the claim that this is enough evidence “of a two-component order parameter”.

The scenario proposed by the authors based on strain is coherent, but it is just a

scenario, there is no direct evidence to support it, only indirect evidence. Indeed, it all rests on the affirmation that “The rings in our study, however, are cut from a single crystal and we do not find any evidence for grain boundaries.” Some kind of microdiffraction measurements of local strain could maybe provide evidence but it is not in the manuscript.

We thank the referee for raising this point. The rings used in this study are all covered with a protective SiO₂ layer that prevents diffraction measurements. We now fabricated a ring that is not covered with SiO₂, using exactly the same FIB parameters. We used Electron Backscatter Diffraction (EBSD) to measure locally the crystal structure along the entire ring circumference. We find the same orientation for the entire ring. While this experiment can not rule out the possibility of some structural defects it does show that there is a single crystal “core” along the entire ring circumference. We added the EBSD data to the supplementary materials.

And at the same time the proposed scenario based on strain has some caveats, for instance:

-The authors themselves also recognize at the end that for their proposed scenario: “However, the fact that this T_c enhancement is only seen with an inplane field is not naturally explained by this scenario.”

-In addition, “The origin of such dislocation defects is not currently known. [...] It is also not clear why the defects occur in roughly half of the devices.” But such a ratio is straightforward in the polycrystalline scenario that the authors dismissed (assuming there is a random number of pi-junctions in the device, that random number can be even or odd, hence a ratio of pi-ring to 0-ring that is close to 50/50).

Before addressing the question of the referee we first comment that in the revised manuscript we put much less emphasis on the microscopic scenario leading to the half-vortex. Indeed, the dislocation scenario is highly unsubstantiated. Next, we emphasize that dislocations are not essential to create a half-vortex of strain. For example, any external source of strain (e.g. from the pads and substrate) can induce non-trivial configurations of strain. Given that the topological classification of such

strain fields is either integer or half-integer it also naturally leads to the 50-50 situation observed in experiment if we assume the strain field is random. Finally, regarding the polycrystalline scenario. As described above we have now added EBSD data, which contradicts the existence of polycrystals in our nano-rings. Moreover, as shown by Ref 22 [Geshkenbein, Larkin, and Barone, PRB 36, 235 (1987)], the polycrystalline scenario for a π -shift still requires a non-s-wave order parameter. If this order parameter does not belong to a multi-dimensional manifold it must have nodes. However, other experiments (STM, specific heat) are in-consistent with nodes. Thus, we deem the polycrystalline scenario unlikely.

By comparison, there is a similar finding in Ref[21] Science 366, 238 (2019) and its supplementary materials, where Y. Li, et al. also find a π shift in the Little-Parks oscillations of rings of Bi₂Pd. However, Y.Li et al. go into much more specific evidence: they find a ratio of eleven π -rings to three 0-rings (or 13 to 8, including some failed devices) only in polycrystalline samples. While in epitaxial sample (with no grain boundaries) they do not find π -rings. They also back it up with TEM cross-section results showing grain boundaries and the size of the grains. Thus, they can quite conclusively attribute the π -shift/half quantum vortex to π -junctions at grain boundaries (thus supporting a p-wave SC scenario). In comparison, I find that the current manuscript is lacking such a similar microstructural study to really back up its claim of a “two-component order parameter”.

We believe our new EBSD data rules out the possibility that our samples contain many grains.

Another thing that gives me doubts is the two-step superconducting transition. I understand it could be due to strain, but, for such a tiny samples, how do you rule out intercalation/contamination by Gallium ions from the FIB cuts ? or maybe even self-doping/intercalation from redeposited materials from the FIB cuts ? or indeed a polycrystalline scenario ? Here comparison with rings fabricated by etching with a noble gas (Helium FIB or Argon ion milling,...) would provide more conclusive evidence.

Also, in the dichalcogenide family, and TaS₂ in particular, there can be some strong effects of disorder on the superconducting T_c (both up or down) due to the competition with the CDW. What if the change in T_c is due to disorder created by the Ga-ion beam or the electron beam of the microscope?

Indeed our samples show two SC transitions or a significant broadening. We do not know the origin of the lower transition but we are certain it represents the transition of the ring. Ga contamination is a reasonable explanation for the reduced T_c of the ring. The change in T_c is rather small (about 300mK). In a trivial superconductor Ga intercalation or disorder in general, can not explain the pi-shifts unless macroscopic grains are formed. We believe that our EBSD results rule out a scenario in which the crystal is broken into pieces with different orientations by the FIB.

Finally, some of the coauthors of this manuscript actually disagree too that this is definitive evidence “of a two-component order parameter”. Indeed they provide an alternative explanation for the pi-shift in ref. M. H. Fischer, P. A. Lee, J. Ruhman arXiv:2304.10583v2 where they even explicitly state so: “[...] this paper offers an alternative explanation of the pi-phase shift to that given in the original paper [3] ...” . So, at least, to ensure self-consistency, the manuscript should be updated to reflect the fact that some of the coauthors proposed an alternative explanation in arXiv:2304.10583v2. I understand this alternative explanation came after the original manuscript was written, but it does rebut the main claim of the title so it cannot stay as is.

In the revised manuscript we have toned down the level of decisiveness of our claims. Indeed, we thought about this for a long while and could not find alternative scenarios. The publication the referee refers to came much later. However, one should note that even this scenario requires quite a lot of coincidence. In particular, it requires magnetic moments to modify the interlayer Josephson coupling to prefer sign-change between H and H' conspiring together with a screw dislocation that happens to go all the way around the ring. Thus, we still argue that

1. The two-component order parameter scenario is the most natural one, and it also explains many other observations in 4Hb-TaS₂.

2. In the absence of crystallographic defects of any kind we can not think of any other scenario but the one including a two component order parameter.

Indeed, the title “Evidence of a two-component OP ...” seems too strong as it is only one possible scenario. “Evidence of half-quantum flux” or “evidence of π -shift in the Little-Parks oscillations” seems more likely to stand the test of time (the current title appears to have already lost that test), all the while being peculiar enough to warrant publication.

So, I would indeed be willing to accept this manuscript, provided the authors strongly requalify their claims, as mentioned above.

We accept the referee's advice. We toned down our claims and changed the title and the text to reflect that.

Report of Referee C:

Almoalem et al. report unusual behaviors in the Little-Parks effect of nanofabricated superconducting 4Hb-TaS₂ rings. First, they find that some of their rings show Little-Parks oscillations with a peak in resistance at zero magnetic field, instead of a dip as in the conventional case. Such a π -shift of the Little-Parks oscillations was previously seen only in a polycrystalline Bi₂Pd ring. Bi₂Pd is a candidate material for topological superconductivity with unconventional pairing and the randomly distributed grains in a polycrystalline ring can allow the occurrence of odd number of π -junctions. It is in this setting that the observation of a π -shift is quite unexpected in a single crystalline ring. The authors proposed that strain in such a single crystal could give rise to an effective “grain boundary”. Second, the authors report that upon applying a small in-plane magnetic field, the resistance of the samples at certain temperatures first decreases and then increases with the perpendicular magnetic field. The authors attribute this decrease to enhanced superconductivity and relate this phenomenon to chiral superconductivity.

While both of the reported experimental features are interesting, the paper seems to lack a comprehensive and in-depth investigation of them.

The important claims such as two-component superconductivity and chiral superconductivity are based on a few traces at a handful of temperature points or in-plane magnetic fields. The authors swiftly ignored other (mundane) origins and jumped directly to the exotic one.

We are pleased that the referee finds our findings interesting.

Following the criticism raised by the referees we toned down our claims in the revised manuscript. Nevertheless, we don't think there is a "mundane" explanation to our observations. Even if, for some unknown reason, our samples contain extended defects our results will imply an unconventional order parameter with nodes. This was so far demonstrated only in a few superconductors. Unconventional superconductivity in a TMD is by itself a very important result.

The manuscript fails to meet the high standard of Nature Communications and I cannot recommend for publication. I list more detailed reasons below:

1. The authors use focused ion beam to fabricate all their rings. This technique typically creates an amorphous layer of about 10-20 nm under grazing incidence polishing [P. Moll, Annu. Rev. Condens. Matter Phys. 9, 147 (2018)]. Both the inner and outer sides of the ring after FIB would host such amorphous layers. Since the thickness of the ring—the difference between the inner and outer radii—is about 100 nm, that means about 20-40% of the ring consists of damaged layers. This damaged layer seems to account for the reduced superconducting transition temperature with a broader transition width, as shown in Fig. 1 and Fig. S4. The presence of a large portion of damaged layers speaks against the authors' claim that their sample is fully single crystalline.

In the revised manuscript we now include an Electron Backscatter Diffraction (EBSD) mapping of a ring prepared without the SiO₂ protective coating. At 10KeV, we estimate the volume from which the electrons are collected to be around 100X100X100 nm³. The inverse pole figure indicates that all the EBSD images show the same orientation. We also map a small region of the crystal far from the

ring part; it also shows the same orientation. The picture emerging is that most of the volume of the rings is the same single crystal as the bulk. Although, we can not rule a thin layer of amorphous material. The EBSD maps and analysis are now added to the supplementary material.

The amorphous layer may consist of polycrystalline domains. These domains can form an odd number of grain boundaries, similar to a tri-crystal of d-wave superconductors. This scenario could explain the occurrence of pi-junctions.

While we agree with the referee that an amorphous layer may form at the edges, we doubt that it can explain the appearance of a pi-junction. The pi-flux in a tri-crystal with a single-component order parameter (with nodes) requires at least three grains with a well defined phase. It is hard to see how this can happen for a SC with a 35nm coherence length in a 10-20nm amorphous layer, especially when it has a single-crystal core. Furthermore, as the referee mentioned, we know that the Ga is lethal for superconductivity and as a result we may assume that the amorphous layers will not be functional, i.e. that part will not superconduct. Finally, as shown by Ref 22 [Geshkenbein, Larkin, and Barone, PRB 36, 235 (1987)], the polycrystalline scenario for a pi-shift still requires a non-s-wave order parameter. If this order parameter does not belong to a multi-dimensional manifold it must have nodes. However, other experiments (STM, specific heat) are in-consistent with nodes. Thus, we deem the polycrystalline scenario unlikely.

In fact, some latest experiments showed that the influence of FIB on two-dimensional materials is much deeper and subtler [F. Sarcan, et al., npj 2D Mater. Appl. 7, 23 (2023)]. As a suggestion, the author may consider using standard e-beam lithography and reactive ion etching to avoid the damage brought by FIB.

Repeating the experiment with standard e-beam lithography fabricated rings is a new project that we hope to perform in the future. This will require using much thinner flakes. Reactive etching can introduce disorder as well.

After the submission of our work, a different group posted a pre-print on arxiv 2302.05078, showing pi-shifts in chiral-molecules intercalated 2H-TaS₂. The rings in their work were fabricated using e-beam. Although the origin of the pi-shifts could be

different, this work suggests that the specific disorder created by the FIB does not play an essential role.

2. The authors spend a substantial portion of their analysis, including Fig. 1 and Fig. 2, on the Little-Parks oscillations of a conventional ring. The readers can only find a limited set of data from the pi-ring (To be exact, just one trace in the main text). Often the readers have to delve into the supplementary information. The authors need to clarify the following issues:

a. Have the authors checked the temperature dependence of ΔR in the pi-rings? Does it agree with that expected from the formula of Little-Parks effect? It is not convincing enough, as is done in the current version, to compare ΔR from experiment and theory at a single temperature point. The authors should present the experimental and theoretical curves of ΔR as a function of T . They should also provide the complete data set of Little-Parks oscillations in all the pi-rings at different temperatures.

The introduction to the Little-Parks effect and a few tests are indeed presented using data from a 0-ring. For this specific sample we measured an extensive set of data for this purpose. We include similar data for a pi-ring in the supplementary. The temperature dependence was shown merely to show that the oscillations are indeed Little-Parks oscillations. The temperature dependence itself has no importance when discussing the symmetry of the order parameter.

In the supplementary material we show the Little-Parks oscillations at different temperatures for a pi-ring as well (Fig. S8).

We added to the supplementary material a calculation of the predicted amplitudes at several different temperatures for both a 0-ring and a pi-ring and these are compared with the data.

b. Apart from the temperature dependence, do the oscillation period agree with the ring geometry in all their rings? It seems that the third pi-ring [Fig. S4(8)] has some double-peak structure around ± 40 Oe. What is the reason for that?

The oscillation period matches the ring geometry in all our rings. In the third pi-ring the oscillations amplitude are very small. This feature at +/- 40Oe is about 2 milliohm, of the order of our noise level, so we think it is statistically irrelevant.

c. How do the pi-rings look like? The authors should provide SEM images for all their rings.

All of our rings have basically the same shape. Only the size varies. We added SEM images of a few more rings, including a pi-ring. Unfortunately, we do not have SEM images for all the rings.

d. Do the authors calibrate zero field in the same way for all the pi-rings? If so, they should present the data.

For all the rings the zero field was determined from a symmetric magneto-resistance curve (explained in the text and Fig.2c explicitly). Based on the numbering of Fig. S6:

- 1) For rings 1 and 4 the LP background is steep and the LP period is long enough to allow a clear determination of the zero field from the LP data.
- 2) Data for ring 2 is shown in the main text.
- 3) We added the curves for rings 5,6,7,8 to the supplementary.
- 4) For ring 3 (Ring II in the main text) which is a 0-ring we could not reproduce the data. The zero field was determined from the MR data.

3. The scenario of some kind of grain boundary formed by strain is highly speculative. I have several doubts on this proposal:

a. it is understood that FIB patterned structures may experience strain. However, that happens for samples that are glued to the substrate [P. Moll, Annu. Rev. Condens. Matter Phys. 9, 147 (2018)]. Here the sample is exfoliated on the substrate and the bonding between the flake and the substrate is van der Waals bonding. It seems unlikely that strain can be created.

We never claimed that strain creates grain boundaries in our rings, on the contrary, we believe there are no grain boundaries in our samples.

In our devices the flakes are covered by a relatively thick layer of SiO₂ that can create strain. In addition the flakes are held in place by the large contacts. So we expect some strain in the rings.

As shown in the supplementary part even a single dislocation can effectively create a pi-junction along the ring.

We emphasize that this is mainly an experimental work, the strain is merely proposed as a possible scenario to explain the pi-shift. In the revised manuscript we removed the discussion about the possible microscopic origin of the topological strain field in the ring. Namely, we do not discuss the existence of dislocations in the main text.

b. The authors show no experimental evidence for the strain effect. In principle, if it is due to strain, there are several ways to check this. For instance, one could do further patterning on the pi-ring to check if it becomes a 0-ring due to the release of strain. A more desirable experiment would be to tune the strain in-situ.

We thank the referee for these suggestions. Controlling the strain is definitely an interesting idea that is worth trying, but it is beyond the scope of the current work.

4. The authors argue that enhanced superconductivity with an in-plane magnetic field stems from chiral superconductivity. In principle, this effect should apply to a bulk superconductor of 4Hb-TaS₂ as well. Have the authors checked this?

The change in T_c extracted from the LP and the change in envelope shape of the MR is similar and of the order of 2-3mK. This is well below the resolution of an R vs T measurement with a transition width of about 200mK. In fact, this is an advantage of the LP effect, it enables us to see very small variations in T_c.

5. Related to the question above, the authors mentioned that their initial intention to introduce the in-plane magnetic field is to stabilize the pi-junction because theory expects that a chiral superconductor should host this property. However, it seems

that they fail to realize this behavior. This negative result seems to question the validity of chiral superconductivity in this system.

We thank the referee for the opportunity to clarify this point. Indeed our initial motivation was to observe half-flux vortices. The LP oscillations period we find, with or without in-plane field, always corresponds to full-flux vortices. Chiral superconductors can support half-flux vortices but also full-flux vortices. The energetic cost of the spin currents associated with half-flux vortices might prevent their presence in our samples, but this does not rule out the possibility of chiral superconductivity in 4Hb-TaS₂ in general.

6. The authors presented only two traces in the main text for the decrease of resistance with increasing magnetic field [Fig. 4(a), Fig. 4(d)]. How does this behavior depend on temperature, as long as the ring becomes superconducting? In fact, the observation that such a w-shaped curve only occurs if the ring becomes superconducting is worrisome. It indicates again that it has something to do with the damaged surface after FIB, as I explained in the first point.

First, we did not measure the effect of an in-plane field at different temperatures, but we do know the W-shaped background is seen only with LP oscillations.

Second, we are puzzled by the comment of the referee about the fact that the W-shape is only observed in superconducting rings. The proposed origin of this shape is an increase in T_c . How can we discuss an increase in T_c if the rings are not superconducting?

The W-shape background only exists when there is a closed superconducting path going around the ring. This closed path is evident by the period of the oscillations that corresponds to the ring area with the correct flux quantization.

If this was not the case, the increase could not be interpreted as an increase in T_c .

We do not see a connection between the W-shaped background and a damaged surface layer.

Furthermore, how does the behavior evolve with the in-plane magnetic field? For the

two traces in Fig. 4(a) and Fig. 4(d), the authors used different in-plane magnetic fields. Why don't the authors study this behavior in a single sample but at a wide range of in-plane magnetic fields?

In the supplementary material we show the in-plane field dependence for one ring. We only scanned a small range of magnetic fields and within this range we don't observe a change in the minimum field of the w-shape.

We hope to conduct a more systematic study of the field and temperature dependence of the T_c enhancement in the future.

7. A home-built coil was used for generating the in-plane magnetic field. It is necessary to explain in more detail how the coils are built: diameter of the coils, distance between the coils, number of turns and layers, thickness of the wire, calibration of the field, simulation of the homogeneity of the field, etc.

We added the requested information to the Methods section:

The Quantum Design PPMS system allows only the application of magnetic fields normal to the sample. To add an in-plane magnetic field we built a Helmholtz coil that is inserted into the PPMS sample space. The coil was designed so the sample is located at its center while inside the PPMS system. The apparatus is made out of two identical coils, with a radius of 10 mm. Each coil contains by 200 turns of a superconducting Sn-Nb wire, with a critical temperature of 7 K, around an aluminum core. We built the apparatus such that the centers of the coils are 10 mm away.

We calibrated the coils outside of the PPMS system using a Tesla-meter, to find a ratio of 180G per 1A. This was compared to a simulation giving the same results.

Inside the PPMS, due to the cooling-power limit of the system, the coils can provide an in-plane field of up to 500 G.

8. The authors argued that the enhancement of superconductivity by an in-plane magnetic field in the previous cases was different because 2H-TaS₂ has strong Ising superconductivity. In principle, Pb has a very strong spin-orbit coupling too. This is evidenced by its large in-plane upper critical field [T. Sekihara, et al., Phys. Rev. Lett. 111, 057005 (2013)], similar to Ising superconductivity. As the authors mentioned,

enhanced superconductivity by an in-plane field was observed in Pb films too. It seems that it is not necessary to involve chiral superconductivity to see this effect. In general, the enhanced superconductivity by an in-plane magnetic field can have multiple origins and remains not fully understood.

We apologize for the confusion. In our rings the enhancement of the transition temperature is measured as a function of the **out-of-plane** field, in the presence of a fixed **in-plane** field. We conjecture that the in-plane field is important to modify the field-coupling of the two-component order parameter such that the weak enhancement becomes visible.

This scenario is different from the enhancement observed in ultra-thin Pb layers, which is measured as a function of an in-plane field. We clarify this point in the text in the revised manuscript.

9. What are the thicknesses of the exfoliated flakes used here?

All of our flakes are 120-150 nm thick.

More specifically, and we use here the notation of Fig. S7:

Sample 1: 125 nm.

Sample 2: 140 nm.

sample 3: 150 nm.

Sample 4: 130 nm.

Sample 5: 150 nm.

Sample 6: 130 nm.

Sample 7: 140 nm.

Sample 8: 150 nm.

10. I recommend the authors to use H_{\perp} instead of H to label the x-axis. Because H may be interpreted as the total magnetic field, instead of the out-of-plane field.

We thank the referee for this suggestion, and indeed we replaced H with H_{\perp} where both in-plane and out-of-plane fields are applied.

(Fig. 4 of the main text, and Fig. S9 in the supplementary material.)

11. The paper seems to be prepared in a hasty manner. Most of the sections in the supplementary information are merely repeating the same texts in the method section. The page number of ref. 42 is wrong.

We have revised the supplementary materials and hope it is now clearer and not repetitive. We note that the methods section is less than 1 page long, the supplementary materials section is 16 pages long. It contains much more information both on the subjects summarized in the Methods sections and on other subjects. We corrected the page number of this reference.

REVIEWER COMMENTS

Reviewer 1 (Remarks to the Author):

I thank the authors for the responses and elaboration on key issues. Having read their report, I still consider the work worthy for a publication in Nature Communication. Concerning the new report, I have two remarks/questions.

Remarks and questions:

- In Fig. S9 you measure the resistivity vs. out-of-plane magnetic field for three values of in-plane fields at a given temperature. How do you explain that the resistivity offset, say $R[H_{\perp} = 0]$ does not evolve monotonously as you increase the in-plane field ?
- Concerning the in-plane magnetic field coupling, I mentioned before. The contribution I had in mind is

$$F = \int dr \int d\gamma \int dz r \boldsymbol{\eta}^{\dagger} \left\{ \alpha_0 B_{\parallel}^2 [\cos(2\varphi_H) \tau^z + \sin(2\varphi_H) \tau^x] \right\} \boldsymbol{\eta}, \quad (0.1)$$

which should be allowed by symmetry. You will find the similar contribution if you use the vector potential $\mathbf{A}(\mathbf{r}) = -\frac{1}{2} \mathbf{r} \times \mathbf{B}$, with $\mathbf{B} = (B_{\parallel} \cos \varphi_H, B_{\parallel} \sin \varphi_H, B_z)$ [i.e. $A_{x,y}$ contain zB_z]. Following that derivation, the prefactor $\alpha_0 \sim z^2$ would be proportional to z^2 —its origin however could also be different.

The contribution (0.1) leads to the B_{\parallel} field-dependence mentioned earlier. I still wonder if this might help to verify/falsify your explanation regarding the T_c -enhancement.

Reviewer #2 (Remarks to the Author):

The authors have properly answered to my remarks. I recommend publication of this manuscript.

Reviewer #3 (Remarks to the Author):

In the revised manuscript and the rebuttal letter, the authors have added some important pieces of information regarding the geometry of the samples, calibrations of the home-built coil, determination of zero field, etc. While I appreciate these clarifications, doubts remain on the two most important points of this paper: the origin of the pi-shift and the mechanism for the w-shaped magneto-resistance. I therefore still cannot recommend the paper for publication.

The first issue is about the presence of 20-40% of amorphous materials in the patterned ring. My major concern is that this amorphous layer could form a polycrystalline ring that may be superconducting, which the authors cannot rule out, and act in a similar way to the polycrystalline Bi2Pd ring that was used to show the pi-shift. I agree that this scenario would still indicate an unconventional order parameter related to TaS₂, or Ga-bombarded TaS₂ to be more precise. But the mechanism is very different from the one proposed in the present paper. The authors showed EBSD data but the resolution, as they stated, is 100 nm. This is comparable to the thickness of their ring—(R_{out}-R_{in}). It is not clear why the EBSD with this limited resolution can rule out the contribution from the amorphous layer. The authors provided a new reference: arXiv 2302.05078 which employs rings fabricated by e-beam lithography. However, their rings have no pi-shift without intercalating chiral molecules. This is compatible with the possibility that the single crystalline core in the FIB patterned ring shows no pi-shift, and the pi-shift stems only from the amorphous layer.

Secondly, the authors argued that the w-shaped magnetoresistance in a static in-plane B cannot be observed in bulk samples due to experimental limitations. Let me break this question into two: (1) do they theoretically (in a thought experiment) expect to see the w-shaped magnetoresistance in a bulk sample of 4Hb-TaS₂? Enhanced superconductivity due to an in-plane magnetic field should be a general feature, irrespective of nano-patterning. I am confused by their answer that “it (the LP effect) enables us to see very small variations in T_c”. Essentially it is the envelop of their theoretical curve that matters, not the oscillations: connecting all the local maxima/minima in Fig. 4 (e) could also show the inverted w-shaped behavior.

(2) In practice, their current samples are already thick (125-150 nm) enough to be in the bulk limit. From the transport data as shown in Fig. 1, there are two transitions: pad transition (red shaded) and ring transition (blue shaded). My former question (point 6) was about the pad transition: do they observe the w-shaped magnetoresistance in the temperature regime where the pad shows superconducting transition? If chiral superconductivity exists over the complete sample, the pad region should also show enhanced superconductivity under an in-plane magnetic field. The sensitivity seems not an obstacle. Here, the resistance drops by 5 Ohm within 50 mK, i.e. $dR/dT \sim 100$ Ohm/K. An enhancement of T_c by 2 mK would correspond to 0.2 Ohm, which is much larger than the amplitude of the detected oscillations.

In fact, there could be a mundane mechanism for the w-shaped magneto-resistance: the in-plane magnetic field enhances the anisotropic ratio between the out-of-plane and in-plane resistivity components: ρ_c/ρ_{ab} . Ramping up the perpendicular magnetic field competes with this trend by enhancing the in-plane resistivity ρ_{ab} . This competition can cause current redistribution in the c-axis such that the measured magnetoresistance shows seemingly non-monotonic behavior. One example for current redistribution caused by varying ρ_c/ρ_{ab} is discussed in Phys. Rev. Lett. 69, 522.

The authors also seem to ignore the multi-layer nature of TaS₂. Applying an in-plane magnetic field can induce Josephson vortices, which are not discussed in the present work. Their theoretical model treated the whole system as a uniform superconductor in the c-axis. A more proper model

should start with the Lawrence-Doniach formula.

Without clarifying these issues above, the current presentation of the w-shaped magnetoresistance seems too preliminary to be attributed to enhanced superconductivity.

Dear Referees,

We thank you for your careful reading of the revised manuscript. In the enclosed document we respond point-by-point to your comments.

Sincerely,

Amit Kanigel on behalf of all authors

Color coding:

Black - original Referee report

Blue - response to Referees

Report of Referee 1

I thank the authors for the responses and elaboration on key issues. Having read their report, I still consider the work worthy for a publication in Nature Communication. Concerning the new report, I have two remarks/questions.

Remarks and questions:

- In Fig. S9 you measure the resistivity vs. out-of-plane magnetic field for three values of in-plane fields at a given temperature. How do you explain that the resistivity offset, say $R[H_{\perp} = 0]$ does not evolve monotonously as you increase the in-plane field ?

The data in the 3 panels in Fig. S9 follows, more or less, the measured temperature, 10mK is a significant difference (the fourth panel is from a different sample). If we compare the two measurements at 2.57K we do find a higher resistance at higher fields.

Saying that, we note that the uncertainty in the sample temperature is larger in the presence of in-plane fields.

As stated in the Methods section, the in-plane field is limited to 500G due to heating problems. Above

500G we find the sample temperature can no longer be controlled. However, we anticipate the heating issue affects the temperature of the sample even for lower in-plane fields.

The temperature in the sample volume of the PPMS system rises due to the heat produced in the in-plane magnetic coil's contacts. This pushes the PPMS away from its normal working conditions. Therefore, the actual sample temperature is expected to slightly deviate from the measured value even though the temperature can be controlled and the field is below 500G. Since we are working at the superconducting transition temperature, any slight change in the temperature can result in a significant change in the resistivity.

- Concerning the in-plane magnetic field coupling, I mentioned before. The contribution I had in mind is

$$F = \int dr \int d\gamma \int dz r \eta^+ \{ \alpha_0 B_{\parallel} [\cos(2\varphi H) \tau^Z + \sin(2\varphi H) \tau^X] \eta \}$$

which should be allowed by symmetry. You will find the similar contribution if you use the vector potential $\mathbf{A}(\mathbf{r}) = -\frac{1}{2} \mathbf{r} \times \mathbf{B}$, with $\mathbf{B} = [B_{\parallel} \cos(\varphi H), B_{\parallel} \sin(\varphi H), B_z]$ [i.e. $A_{x,y}$ contain zB_z]. Following that derivation, the prefactor $\alpha_0 \sim z^2$ would be proportional to z^2 —its origin however could also be different. The contribution leads to the B_k field-dependence mentioned earlier. I still wonder if this might help to verify/falsify your explanation regarding the T_c -enhancement.

We apologize for the misunderstanding. Indeed, this term is allowed and will lead to a dependence of T_c on the in-plane magnetic field. As a side remark, this coupling could be used to verify the general assumption that there is a two-component order parameter in 4Hb-TaS2 in general. For example, T_c is expected to exhibit a “two-fold” dependence on the in-plane magnetic field. In fact, this effect was measured by the authors of arXiv:2208.14442, where the axis of η was assumed to be fixed by strain.

Regarding our experiment; we agree with the referee that the above coupling can be exploited to strengthen/weaken our claims. Because the term has a very similar effect to strain (it prefers the nematic order parameter over the chiral one), we could imagine that when the in-plane field is collinear with strain the energy of the nematic term is reduced, and therefore it becomes even harder for the chiral term to compete with the nematic component. On the other hand, when the in-plane field and strain are perpendicular, the nematic component is frustrated, and thus elevated in energy. In this case, the chiral term will have a larger effect. According to this picture, the T_c enhancement should be largest at the value of φ_H where T_c is minimal and vice versa.

Report of Referee 2

The authors have properly answered to my remarks. I recommend publication of this manuscript.

We express our gratitude to the referee for thoroughly reviewing our paper and for providing a favorable assessment of our research.

Report of Referee 3

In the revised manuscript and the rebuttal letter, the authors have added some important pieces of information regarding the geometry of the samples, calibrations of the home-built coil, determination of zero field, etc. While I appreciate these clarifications, doubts remain on the two most important points of this paper: the origin of the pi-shift and the mechanism for the w-shaped magneto-resistance. I therefore still cannot recommend the paper for publication.

As a general remark, we emphasize that this is first and foremost an experimental paper. Our data provides undoubtable evidence for pi-shifts in the Little-Parks oscillations and unconventional field dependence of the magneto-resistance in the presence of an in-plane magnetic field.

These two results, that the referee mentioned as the most important part of the work, are clearly shown to exist in several samples under different experimental conditions. They provide clear evidence for unconventional superconductivity, as the referee pointed in the report.

To explain these observations, we suggested a scenario that we believe to be a reasonable explanation to the experimental observations. We understand that the referee still has doubts regarding our theoretical explanations, but we insist that the main justification for publication should be the data and not its presumed origin.

The first issue is about the presence of 20-40% of amorphous materials in the patterned ring. My major concern is that this amorphous layer could form a polycrystalline ring that may be superconducting, which the authors cannot rule out, and act in a similar way to the polycrystalline Bi₂Pd ring that was used to show the pi-shift. I agree that this scenario would still indicate an unconventional order parameter related to TaS₂, or Ga-bombarded TaS₂ to be more precise. But the mechanism is very different from the one proposed in the present paper. The authors showed EBSD data but the resolution, as they stated, is 100 nm. This is comparable to the thickness of their ring—(R_{out}-R_{in}). It is not clear why the EBSD with this limited resolution can rule out the contribution from the amorphous layer. The authors provided a new reference: arXiv 2302.05078 which employs rings fabricated by

e-beam lithography. However, their rings have no pi-shift without intercalating chiral molecules. This is compatible with the possibility that the single crystalline core in the FIB patterned ring shows no pi-shift, and the pi-shift stems only from the amorphous layer.

We don't know if 20-40% of the sample is amorphous. On the other hand, based on the EBSD data, we do know that 60-80% of the sample is a well oriented single crystal. The referee completely ignores that for some reason.

The referee proposes an alternative scenario, a **single component** non-s-wave order parameter that is frustrated by polycrystalline domains. However, an amorphous crystal is very different from a polycrystalline one. It is not expected to lead to sharp transitions between crystallographic orientations which are essential to frustrate such an order parameter. Moreover, non-s-wave order parameters are sensitive to disorder and therefore the high sensitivity of TaS₂ to Ga-poisoning strongly suggests that the disordered edges will not be superconducting. If we assume the amorphous part is superconducting, surely it will have a lower T_c compared to the single crystalline part. In this case, the single crystal core that has a higher T_c will shortcut the disordered ring and there will be no Little-Parks oscillations at the T_c of the amorphous part.

We don't think the scenario proposed by the referee is more reasonable than the scenario proposed in the manuscript. Nevertheless, we did admit in the revised manuscript that we can not rule out the existence of extended defects. For that reason we toned down our claims and left the two-components superconductivity merely as a possible explanation.

Secondly, the authors argued that the w-shaped magnetoresistance in a static in-plane B cannot be observed in bulk samples due to experimental limitations. Let me break this question into two:

(1) do they theoretically (in a thought experiment) expect to see the w-shaped magnetoresistance in a bulk sample of 4Hb-TaS₂? Enhanced superconductivity due to an in-plane magnetic field should be a general feature, irrespective of nano-patterning. I am confused by their answer that "it (the LP effect) enables us to see very small variations in T_c". Essentially it is the envelop of their theoretical curve that matters, not the oscillations: connecting all the local maxima/minima in Fig. 4 (e) could also show the inverted w-shaped behavior.

(2) In practice, their current samples are already thick (125-150 nm) enough to be in the bulk limit. From the transport data as shown in Fig. 1, there are two transitions: pad transition (red shaded) and ring transition (blue shaded). My former question (point 6) was about the pad transition: do they observe the

w-shaped magnetoresistance in the temperature regime where the pad shows superconducting transition? If chiral superconductivity exists over the complete sample, the pad region should also show enhanced superconductivity under an in-plane magnetic field. The sensitivity seems not an obstacle. Here, the resistance drops by 5 Ohm within 50 mK, i.e. $dR/dT \sim 100$ Ohm/K. An enhancement of T_c by 2 mK would correspond to 0.2 Ohm, which is much larger than the amplitude of the detected oscillations.

The proposed origin for the “W” shape exists also in the bulk. However, we do not expect the “W” shape to appear in the magneto-resistance of bulk samples. In the bulk there are additional sources for magneto-resistance, such that one can not simply relate ΔR to ΔT_c . Making such a connection requires a continuous SC path along the entire ring. Indeed, the fact that the “W” is seen only together with the oscillations, which indicate the conditions for the Little-Parks effects are met.

Therefore, we do expect an increase in T_c in the presence of an in-plane field at small out-of-plane fields in bulk 4Hb-TaS₂. The expected increase is of the order of 5mK, an increase that experimentally can not be observed in a R vs. T curve. However, the magneto-resistance at a fixed temperature (not in the Little-Parks regime) will not necessarily show the increase in T_c .

In fact, there could be a mundane mechanism for the w-shaped magneto-resistance: the in-plane magnetic field enhances the anisotropic ratio between the out-of-plane and in-plane resistivity components: ρ_c/ρ_{ab} . Ramping up the perpendicular magnetic field competes with this trend by enhancing the in-plane resistivity ρ_{ab} . This competition can cause current redistribution in the c-axis such that the measured magnetoresistance shows seemingly non-monotonic behavior. One example for current redistribution caused by varying ρ_c/ρ_{ab} is discussed in Phys. Rev. Lett. 69, 522.

The paper the referee is referring to deals with flux-flow. We do not think the W-shape in the magneto-resistance is related to flux-flow for several reasons:

- 1) Flux-flow depends strongly on the magnitude of the current. We work at very low current densities. And we show in the supplementary materials that the magneto-resistance does not depend on the current.
- 2) As shown in S9, the w-shape seems to depend weakly on the value of the in-plane field.
- 3) Most importantly, as we already mentioned, the W-shape is observed only concurrently with the Little-Parks oscillations. This can not be explained by flux-flow.

The authors also seem to ignore the multi-layer nature of TaS₂. Applying an in-plane magnetic field can induce Josephson vortices, which are not discussed in the present work. Their theoretical model treated the whole system as a uniform superconductor in the c-axis. A more proper model should start with the Lawrence-Doniach formula.

We indeed ignored the role of vortices, based on the arguments provided in the preceding paragraph. Our theoretical model is actually in the extreme limit of a multi-layer superconductor, we assume the sample is a stack of uncoupled superconducting layers. Namely, the GL theory we write is applicable to a single two-dimensional layer.

Without clarifying these issues above, the current presentation of the w-shaped magnetoresistance seems too preliminary to be attributed to enhanced superconductivity.

REVIEWERS' COMMENTS

Reviewer #1 (Remarks to the Author):

I thank the authors, once again, for their clarifications and responses, and I continue to recommend the manuscript for publication in *Nature Communications*.

Reviewer #3 (Remarks to the Author):

In addressing my concern on the absence of w-shaped magnetoresistance in bulk TaS₂, the authors argued that bulk TaS₂ have magneto-resistance from additional sources. However, these unspecified additional sources may play a similar role in the FIB patterned rings. Furthermore, the Little-Parks effect itself does not relate ΔT_c to ΔR . It is the superconducting transition regime with a large derivative dR/dT which can convert a small variation in T_c to a measurable change in resistance. In this regard, an increase of $\Delta T_c=5$ mK, by ramping up the perpendicular B in the presence of an in-plane B, should result in a measurable change (about 0.5 Ohm) in magneto-resistance via $dR/dT \cdot \Delta T_c$ inside the TaS₂ flake without the ring pattern.

One possible reason why this change was so far not observed in the authors' experiment may be due to the limited temperature range for this phenomenon to occur. The absence of w-shaped magneto-resistance in Fig. 4(b) only indicates that T_c enhancement does not occur in the bulk of TaS₂ at this particular temperature. It may occur at a lower temperature but cannot be distinguished from the contribution from the ring. Future experiments may help elucidate this uncertainty.

All in all, the observation of pi-shift and w-shaped magnetoresistance in a FIB patterned ring of TaS₂ is unusual and worth reporting.

Dear Referees,

We thank you for your careful reading of the revised manuscript. In the enclosed document we respond point-by-point to your comments.

Sincerely,

Amit Kanigel on behalf of all authors

Color coding:

Black - original Referee report

Blue - response to Referees

Report of Referee 1

I thank the authors, once again, for their clarifications and responses, and I continue to recommend the manuscript for publication in Nature Communications.

We thank the referee for carefully reviewing our work.

Report of Referee 3

In addressing my concern on the absence of w-shaped magnetoresistance in bulk TaS₂, the authors argued that bulk TaS₂ have magneto-resistance from additional sources. However, these unspecified additional sources may play a similar role in the FIB patterned rings. Furthermore, the Little-Parks effect itself does not relate ΔT_c to ΔR . It is the superconducting transition regime with a large derivative dR/dT which can convert a small variation in T_c to a measurable change in resistance. In this regard, an increase of $\Delta T_c = 5$ mK, by ramping up the perpendicular B in the presence of an in-plane B, should result in a measurable change (about 0.5 Ohm) in magneto-resistance via $dR/dT \cdot \Delta T_c$ inside the TaS₂ flake without the ring pattern.

We thank the referee for carefully reviewing our work.

We agree with the referee that whatever affects the magneto-resistance beyond the LP effect should be seen both in the bulk and in patterned samples. In fact, we use this magneto-resistance to find the zero field of our magnet.

We are not sure about the referee's estimate of the expected change in resistance. The resistance at T_c of our flakes (before the ring patterning) is about 0.5 Ohm.

Using the derivative of the resistance with respect to the temperature we expect only a change of about a mOhm per mK change in T_c . This cannot be measured directly.

We could try to make smaller devices with higher resistance, but this would be a completely new study.

One possible reason why this change was so far not observed in the authors' experiment may be due to the limited temperature range for this phenomenon to occur. The absence of w-shaped magneto-resistance in Fig. 4(b) only indicates that T_c enhancement does not occur in the bulk of TaS₂ at this particular temperature. It may occur at a lower temperature but cannot be distinguished from the contribution from the ring. Future experiments may help elucidate this uncertainty.

Indeed, we cannot rule out that the w-shape in the bulk appears at a different temperature. We can only say that in our rings the w-shape was always found in the temperature range where LP oscillations were observed.

Following the referee's suggestion we intend to try to measure the T_c enhancement directly by making smaller bridge-like devices with higher resistance and without the LP oscillations.

All in all, the observation of pi-shift and w-shaped magnetoresistance in a FIB patterned ring of TaS₂ is unusual and worth reporting.

We thank the referee for the fruitful discussion and for finding our results worth publishing.